



# Quantifying the Importance of Vehicle Ammonia Emissions in an Urban Area of the Northeastern US Utilizing Nitrogen Isotopes

Wendell W. Walters[*1,2], Madeline Karod[1,3], Emma Willcocks[4], Bok H. Baek[5], Danielle E. Blum[2,6], and Meredith G. Hastings[1,2]

[1]Department of Earth, Environmental, and Planetary Sciences, Brown University; Providence, RI 02912, USA.
[2]Institute at Brown for Environment and Society, Brown University; Providence, RI 02912, USA
[3]Chemistry and Physics Department, Simmons University; Boston MA 02215, USA
[4]Program in Biology, Division of Biology and Medicine, Brown University; Providence, RI 02912, USA
[5]Center for Spatial Information, Sciences, and Systems, George Mason University; Fairfax, VA 22030, USA
[6]Department of Chemistry, Brown University; Providence, RI 02916, USA

Correspondence to: Wendell W. Walters (wendell_walters@brown.edu)

**Abstract.** Atmospheric ammonia ($NH_3$) is a critical component of our atmosphere that contributes to air quality degradation and reactive nitrogen deposition; however, our knowledge of $NH_3$ in urban environments remains limited. Year-long ambient
$NH_3$ and related species were measured for concentrations and the nitrogen isotopic compositions ($\delta^{15}N$) of $NH_3$ and particulate ammonium ($pNH_4^+$) to understand the temporal sources and chemistry of $NH_3$ in a northeastern US urban environment. We found that urban $NH_3$ and $pNH_4^+$ concentrations were elevated compared to regional rural background monitoring stations, with seasonally significant variations. Local and transported sources of $NH_x$ ($NH_3 + pNH_4^+$) were identified using polar bivariate and statistical back trajectory analysis, which suggested the importance of vehicles, volatilization, industry, fuel
combustion, and biomass burning emissions. Utilizing a uniquely positive $\delta^{15}N(NH_3)$ emission source signature from vehicles, a Bayesian stable isotope mixing model indicates that vehicles contribute 30.7±11.6% (mean±1σ) to the annual background level of urban $NH_x$, with a strong seasonal pattern with higher relative contribution during winter (45.8±13.0%) compared to summer (20.8±9.7%). The decrease in the relative importance of vehicle emissions during the summer was suggested to be driven by temperature-dependent $NH_3$ emissions from volatilization sources based on wind direction, back
trajectory, and $NH_3$ emission inventory analysis. This work highlights that reducing vehicle $NH_3$ emissions should be considered to improve wintertime air quality in this region.

## 1. Introduction

Ammonia (NH₃) is a critical component of the atmosphere and the global nitrogen cycle (Behera et al., 2013; Galloway et al., 2004). As the primary alkaline atmospheric molecule, $NH_3$ plays an important role in neutralizing atmospheric acids, leading to fine particulate matter ($PM_{2.5}$), including particulate ammonium ($pNH_4^+$), which have important implications for air quality, human health, visibility, and climate change (Behera and Sharma, 2010; Updyke et al., 2012; Wang et al., 2015). Agricultural activities, including fertilizer application and livestock waste, dominate the emission of $NH_3$, accounting for over 60% of the

global inventory (Bouwman et al., 1997); however, there are significant $NH_3$ spatiotemporal variabilities due to its short atmospheric lifetime, typically a few hours to a day, and numerous emission sources (Van Damme et al., 2018). Urban regions have been shown to have elevated levels of $NH_3$ and reduced nitrogen deposition (Plautz, 2018; Joyce et al., 2020; Hu et al., 2014; Decina et al., 2020, 2017), indicating the potential for important non-agricultural emission sources that may disproportionately impact human and environmental health. In recent years, quantifying surface-level $NH_3$ and its deposition

products in the US has been a focus of several national monitoring networks, including the Ammonia Monitoring Network (AMoN), the Interagency Monitoring of Protected Visual Environments (IMPROVE), the National Atmospheric Deposition Program (NADP), and the Clean Air Status and Trends Network (CASTNET). However, these measurements are typically conducted in rural locations. Long-term records of $NH_3$ and its deposition products in urban regions are exceedingly scarce, which often leads to models evaluated to observations primarily conducted in rural locations.


The $NH_3$ sources contributing to the urban budget remain contested. Several studies have identified vehicle emissions as a major urban $NH_3$ emission source (Sun et al., 2017, 2014; Suarez-Bertoa et al., 2014, 2017). In contrast, other studies have suggested that vehicle emissions are relatively unimportant for urban regions and instead have found evidence for significant local and transported emissions due to temperature-dependent volatilization sources (Hu et al., 2014; Yao et al., 2013; Nowak

et al., 2006). Recent satellite observations, taking advantage of the COVID-19 lockdown period, have for the first time confirmed vehicle emissions as a significant localized source of $NH_3$ in an urban region (Cao et al., 2021). However, quantifying the contribution of local urban $NH_3$ emissions to the urban background is complex as it is coupled to meteorological



parameters that influence $NH_3$ and particulate ammonium ($pNH_4^+$) partitioning, mixing/dispersion of local emissions, and contributions via long-range transport from agricultural regions (Meng et al., 2011; Walker et al., 2004).


The nitrogen stable isotopic composition ($\delta^{15}N(‰) = [(^{15}R_{sample})/(^{15}R_{reference})-1]\times1000$, where $^{15}R$ is the ratio of $^{15}N/^{14}N$, and air is the N isotopic reference) may be a useful chemical fingerprinting tool to track source contributions and validate model apportionments of urban $NH_3$ (Felix et al., 2017, 2013). Indeed, numerous studies have utilized $\delta^{15}N$ of $NH_3$ and $pNH_4^+$ for source apportionment (Felix et al., 2017; Pan et al., 2016; Berner and Felix, 2020; Liu et al., 2018; Pan et al., 2018; Wu et al.,

2019; Bhattarai et al., 2020; Xiao et al., 2020; Zhang et al., 2021), taking advantage of the suggested lower $\delta^{15}N$ signatures of agricultural $NH_3$ emissions relative to fossil fuel combustion (Felix et al., 2013; Chang et al., 2016). However, it has been shown that collection techniques can play an important role on the measured $\delta^{15}N$ values (Skinner et al., 2006). Passive collection of $NH_3$ has been shown to induce a $\delta^{15}N$ bias relative to active collection techniques due to a potential diffusion isotope effect, such that many of the $\delta^{15}N(NH_3)$ studies may be unreliable (Walters et al., 2020; Pan et al., 2020; Kawashima

et al., 2021; Crittenden et al., 2015). Additionally, collections of $pNH_4^+$ on aerosol filters may be impacted by the loss of semi-volatile $NH_4NO_3$ leading to an undesirable $\delta^{15}N$ fractionation (Walters et al., 2019). Thus, there could be inaccuracies in the previously reported source apportionment results related to the collection technique used to concentrate ambient $NH_3$ and $pNH_4^+$ for offline $\delta^{15}N$ characterization.

An additional complication for utilizing $\delta^{15}N$ for source apportionment is the role of isotope fractionation associated with $NH_3$ and $pNH_4^+$ phase partitioning (Urey, 1947). This phase partitioning is driven by a thermodynamic equilibrium that depends on the relative humidity, temperature, and particle chemical composition and is typically achieved on the order of tens of minutes (Meng and Seinfeld, 1996). This suggests that isotope equilibrium between $NH_3$ and $pNH_4^+$ should be achieved under most ambient atmosphere conditions. Absorption of $NH_3$ in a deliquesced particle can be represented by the following (R1):

$$NH_{3(g)} + H_2O_{(l)} \leftrightharpoons NH_3 * H_2O \leftrightharpoons NH_{4(aq)}^+ + OH^- \qquad \text{R1}$$

The associated $NH_{3(g)}$-$pNH_4^+_{(aq\ or\ s)}$ nitrogen equilibrium isotope exchange has been suggested to be represented through the following reactions (Walters et al., 2018; Urey, 1947) (R2-R4):



$$^{15}NH_{3(g)} + {}^{14}NH_{4(s)}^{+} \rightleftharpoons {}^{14}NH_{3(g)} + {}^{15}NH_{4(s)}^{+} \qquad \text{R2}$$

$$^{15}NH_{3(g)} + {}^{14}NH_{4(aq)}^{+} \rightleftharpoons {}^{14}NH_{3(g)} + {}^{15}NH_{4(aq)}^{+} \qquad \text{R3}$$

$$^{15}NH_{3(g)} + {}^{14}NH_{3(aq)} \rightleftharpoons {}^{14}NH_{3(g)} + {}^{15}NH_{3(aq)} \qquad \text{R4}$$

Recent theoretical calculations predict nitrogen equilibrium constants (K) or isotopic fractionation factors ($\alpha$) of 1.031($\pm$0.004), 1.034($\pm$0.004), and 1.004($\pm$0.003) at 25 °C for R2, R3, and R4, respectively (Walters et al., 2018). These calculations indicate that isotope equilibrium will lead to the preferential partitioning of $^{15}N$ in the condensed phase of $NH_x$ with an isotopic enrichment factor ($\varepsilon$(‰) = 1000[$\alpha$-1]) ranging from 4.0 to 34‰ depending on the equilibrium reaction (Walters et al., 2018).

Previous work has also suggested that unidirectional $NH_3$ neutralization reactions involving sulfuric acid ($H_2SO_4$) may induce a kinetic isotope effect of approximately -28‰ (Pan et al., 2016). These potential competing equilibrium and kinetic isotope effects indicate that the phase $\delta^{15}N$ fractionation between $NH_3$ and $pNH_4^+$ is complex. However, most previous $\delta^{15}N(NH_3)$ source apportionment studies have collected a single phase of $NH_x$ and have either not considered $\delta^{15}N$ fractionation or estimated the effect based on only $NH_3/NH_4^+$ equilibrium exchange reactions (R2 & R3). This may not be a robust approach

due to the wide range of $\delta^{15}N$ fractionation predicted from equilibrium and/or kinetic isotope effects (i.e., -28 to 34‰), leading to inaccurate source apportionment results. Indeed, laboratory dynamic flow chamber experiments and field measurements from simultaneously collected $NH_3$ and $pNH_4^+$ have reported lower $\varepsilon$ than calculated for R2 and R3 (Walters et al., 2019; Kawashima and Ono, 2019). Thus, predicting the isotope fractionation factor associated with $NH_3$ and $pNH_4^+$ phase partitioning under varying ambient conditions remains unclear (Bhattarai et al., 2021), implying that speciated and

simultaneous collections of $NH_x$ is needed to account for the complex $NH_3/pNH_4^+$ fractionation to reduce the influence of phase-dependent $\delta^{15}N$ variabilities.

Improving the reliability of $\delta^{15}N$ source apportionment results of $NH_3$ requires using robust methods suitable for $\delta^{15}N$ characterization that can also account for phase-dependent $\delta^{15}N$ variabilities. In this study, we have characterized the seasonal

ambient $NH_x$ ($NH_3$ + $pNH_4^+$) source contributions using concentration and isotope measurements at an urban site in Providence, RI, US, using laboratory-verified and field-tested collection techniques shown to quantitatively collect $NH_x$ for accurate and precise $\delta^{15}N$ characterizations (Walters and Hastings, 2018; Walters et al., 2019). The study site is a mid-sized



coastal city located within the northeastern US megapolis. This is an important region to monitor because the northeastern US wintertime air quality has not improved as much as expected, despite aggressive reductions of precursor emissions in recent

decades (Shah et al., 2018). We have recently characterized the $\delta^{15}N(NH_3)$ from urban vehicle plumes, which has indicated this source to have a unique positive $\delta^{15}N$ signature of 6.6±2.1‰ compared to other $NH_3$ sources that tend to have negative $\delta^{15}N$ values (Walters et al., 2020). Here we aim to quantify the importance of vehicle $NH_3$ emissions at our urban site. Our study contributes to the first $\delta^{15}N$ measurements of speciated $NH_x$ in New England and contributes to our understanding of seasonal urban $NH_x$ source apportionment in an environment that particulate nitrate ($pNO_3^-$) formation is commonly $NH_3$-

limited (Park et al., 2004).

## 2. Materials, Methods, and Datasets

### 2.1 Collection of $NH_x$ and Associated Gases and Particles

Simultaneous collections of reactive gases and $PM_{2.5}$ were conducted using a series of coated glass honeycomb denuders and

a downstream filter pack housed in a ChemComb Speciation Cartridge. This sampling system has been extensively evaluated for its ability to speciate between inorganic gases and particulate matter for offline concentration determination (Koutrakis et al., 1993, 1988). Additionally, this system is a suitable technique for the characterization of $\delta^{15}N(NH_3)$ and $\delta^{15}N(pNH_4^+)$ with a precision of ±0.8‰ and ±0.9‰ (1σ), respectively (Walters and Hastings, 2018; Walters et al., 2019). Briefly, the sampler consisted of a PTFE-coated inlet to minimize reactive gas loss, a $PM_{2.5}$ impactor plate, a basic-coated honeycomb denuder

(2% carbonate (w/v) + 1% glycerol (w/v) in 80:20 water-methanol (v/v) solution) to collect acidic gases including nitric acid ($HNO_3$) and sulfur dioxide ($SO_2$), an acid-coated denuder (2% citric acid (w/v) + 1% glycerol (w/v) in 20:80 water-methanol (v/v) solution) to collect $NH_3$, and a filter pack consisting of a Nylon and 5% (w/v) citric acid-coated cellulose filter for the collection of $pNH_4^+$. All denuder and filter preparation, handling, and extraction techniques have been previously described (Walters and Hastings, 2018; Walters et al., 2019). The samplers were held vertically to limit the potential for gravitational

settling of particles on the denuder surfaces and were housed in a custom-built weather-protected container. Ambient air was sampled at a flow rate of 10 liters per minute. Collections were conducted for 24 h (15:00 to 15:00 the following day) approximately twice per week in Providence, RI, US (41.83 ºN, 71.40 ºW) on the rooftop of a building from February 6, 2018, to February 1, 2019 (Figure 1). The study location is a mid-sized coastal city within New England, with an approximate population of 180,000 and population density of 3,800 per $km^2$. The monitoring location is in an urban-mixed use region that

includes commercial buildings, residential buildings, highways, and industry with some clear $NH_3$ point sources such as vehicles, residential heating, sewage, and industrial emission.



## 2.2 Concentration and $\delta^{15}N(NH_x)$ Isotopic Analysis

The concentrations of the denuder and filter extraction solutions were analyzed using colorimetry and ion chromatography analytical techniques. The colorimetric analysis included measurements of $[NH_4^+]$ using the indophenol blue method (i.e., US EPA Method 350.1) and $[NO_2^-]$ via diazotization with sulfanilamide dihydrochloride (i.e., US EPA Method 353.2) that was automated by a discrete UV-Vis spectrophotometer (Westco SmartChem). Anion concentrations that included $[Cl^-]$, $[NO_3^-]$, and $[SO_4^{2-}]$ were analyzed using ion chromatography (Dionex DX500). The limit of detection (LOD) of was approximately 0.5 $\mu mol \cdot L^{-1}$ for $[NH_4^+]$ and $[NO_2^-]$ and 2 $\mu mol \cdot L^{-1}$ for $[Cl^-]$, $[NO_3^-]$, and $[SO_4^{2-}]$. The relative standard deviations for all quantified ions were less than 5%. Laboratory blanks of denuder and filter samples were periodically taken, representing approximately 10% of the collected samples. The blanks were below our LOD, except for $[Cl^-]$ that had a large and variable blank for both the carbonate denuder and Nylon filter, such that this data was not reported in this work.

The determination of $\delta^{15}N$ of the $NH_4^+$ in the denuder and filter extracts was conducted using a chemical technique that converts $NH_4^+$ to $NO_2^-$ using an alkaline hypobromite solution and reducing the generated $NO_2^-$ to $N_2O$ using sodium azide in an acetic acid buffer solution (Zhang et al., 2007). The generated $N_2O$ was purified and concentrated using an automated extraction system coupled to a continuous flow Isotope Ratio Mass Spectrometer for $\delta^{15}N$ determination as previously described (Walters and Hastings, 2018). In each sample batch, unknowns were calibrated to two internationally recognized $NH_4^+$ isotopic reference materials, IAEA-N2 and USGS25, with $\delta^{15}N$ values of 20.3‰ and -30.3‰ (Bohlke et al., 1993; Böhlke and Coplen, 1993), respectively. An in-house $NH_4^+$ quality control ($\delta^{15}N = -1.5$‰) and an $NO_2^-$ reference material with a known isotope composition (RSIL-N10219; $\delta^{15}N= 2.8$‰) (Böhlke et al., 2007) were also run intermittently as quality control to monitor the conversion of $NO_2^-$ to $N_2O$ and system stability across runs. Corrections to determine $\delta^{15}N(NH_4^+)$ are performed by accounting for isobaric influences, blank effects, and calibrating the unknowns to the internationally recognized $\delta^{15}N(NH_4^+)$ standards. The correction scheme resulted in an average slope between the measured $\delta^{15}N(N_2O)$ and the standard $\delta^{15}N(NH_4^+)$ values of 0.501 ± 0.024 near the theoretical line of 0.500 for the azide/acetic acid reduction method (Zhang et al., 2007; McIlvin and Altabet, 2005). The pooled standard deviations of the isotopic reference materials were ±0.6‰ (n=62), ±0.7‰ (n=62), ±0.5‰ (n=14), and ±1.3‰ (n=18), for IAEA-N2, USGS25, in-house $NH_4^+$, and RSIL-N10219, respectively. Due to the numerous steps and potential interferences associated with the employed chemical conversion technique, we established the following quality assurance criteria for our sample unknowns: (1) $[NH_4^+]$ greater than 5 $\mu mol \cdot L^{-1}$ to combat the significant alkaline hypobromite reagent blank, (2) $[NO_2^-]/[NH_4^+]$ ratio less than 5% since $NO_2^-$ is an interferent, and (3) quantitative yield of $NH_4^+$ to $NO_2^-$ conversion (i.e., incomplete conversion would lead to undesirable $\delta^{15}N$ fractionation). These criteria were met for 89 out of 97 $NH_3$ samples and 60 out of 97 $pNH_4^+$ samples. Replicate measurements of sample unknowns across batch analyses was conducted for approximately 10% of samples and had an average deviation of ±1.4‰.



170

### 2.3 Ancillary Datasets

Annual emission data of $NH_3$ at the county level was accessed from the US EPA National Emission Inventory 2014 (NEI-14), and chemically speciated gridded hourly $NH_3$ emission data was generated using the Sparse Matrix Operator Kerner Emissions (SMOKE) model (Baek and Seppanen, 2021). The SMOKE processor was initialized using the NEI-2014 emissions modeling platform (EMP) version 7.1, as this was the most recently available NEI at the time of the analysis. The model output was binned by month. Ancillary meteorological parameters were accessed from the Rhode Island Department of Health air monitoring and Chemical Speciation Network (CSN) monitoring station at East Providence (Figure 1). Data were accessed from co-located Ammonia Monitoring Network (AMoN) and Clean Air Status and Trends Network (CASTNET) stations located within New England (US EPA Region 1) for $[NH_3]$ and $[pNH_4^+]$, respectively. These sites included Abington, CT (41.84°N, 72.01°W), Underhill, VT (44.53°N, 72.87°W), Woodstock, NH (44.53°N, 72.87°W), and Ashland, ME (46.60°N, 68.41°W) (Figure 1). Archived back trajectories and boundary layer heights were computed using the NOAA Air Resource Lab HYSPLIT model (Stein et al., 2015). 72-h back trajectories were calculated arriving at Providence, RI (41.73°N, 71.43°W) using the NAM 12 km meteorology initiated at the end of each sampling period. Atmospheric $NH_x$ has a lifetime typically on the order of 2.1 days (Paulot et al., 2016), such that the chosen trajectory time should account for the potential of long-range transport of $NH_x$ to the sampling site. A new back trajectory was calculated every 3 h for a max of 8 trajectories encompassing the 24 h sampling period at 100 m above ground level.

### 2.4 Statistical Analyses

Geospatial statistical analysis that included bivariate wind direction and wind speed polar plots and back-trajectory clustering was conducted using the 'open-air' program package using R (Carslaw and Ropkins, 2012). Local $NH_x$ source identification was estimated using the conditional bivariate probability function (CBPF) analysis that provides a conditional probability field for high concentrations dependent on wind speed and direction (Uria-Tellaetxe and Carslaw, 2014). It is defined as the following (Eq. 1):

$$CBPF_{\Delta\theta,\Delta u} = \frac{m_{\Delta\theta,\Delta u|c\geq x}}{n_{\Delta\theta,\Delta u}} \qquad\qquad (\text{Eq. 1})$$

where $m_{\Delta\theta,\Delta u}$ is the number of samples in the wind sector $\Delta\theta$ with wind speed interval $\Delta u$ having concentration C greater than a threshold value x, $n_{\Delta\theta,\Delta u}$ is the total number of samples in that wind direction-speed interval. The threshold values were set as the top 25% concentration for these analyses. These bivariate polar plots show how a concentration of species varies with wind speed and direction in polar coordinates and are useful in characterizing emission sources (Carslaw and Ropkins, 2012; Carslaw et al., 2006; Tomlin et al., 2009; Zhou et al., 2019). Additionally, source locations that contribute to long-range $NH_x$ transport were evaluated using the potential source contribution function (PSCF). This analysis combines atmospheric



concentrations with air mass trajectories and uses residence time information to identify air parcels that contribute to high concentrations at a receptor site (Fleming et al., 2012; Pekney et al., 2006; Begum et al., 2005). The PSCF calculation indicates the probability that a source is located at latitude $i$ and longitude $j$ and is calculated as the following (Eq. 2):

$$PSCF = \frac{m_{ij}}{n_{ij}} \qquad \text{(Eq. 2)}$$

where $n_{ij}$ is the number of times that the trajectories pass through the cell (i,j) and $m_{ij}$ is the number of times that a source concentration was high when the trajectories passed through the cell (i,j), and the criterion for determining $m_{ij}$ was defined as the 90th percentile (Carslaw and Ropkins, 2012).

## 3. Results and Discussion

### 3.1 Urban $NH_3$ and $pNH_4^+$ Temporal Concentrations

The urban $NH_3$ and $pNH_4^+$ were monitored under a range of meteorological conditions (Figure 2). The annual $[NH_3]$ ranged from 0.234 to 2.94 $\mu g/m^3$ with a mean of 0.890±0.517 $\mu g/m^3$ (n=97), and $[pNH_4^+]$ ranged from 0.019 to 1.62 $\mu g/m^3$ with a mean of 0.412±0.287 $\mu g/m^3$. The $NH_x$ partitioning between gas and particle-phase was quantified as $fNH_3$ ($fNH_3$ = $[NH_3]_{mol}/([NH_3]_{mol} + [pNH_4^+]_{mol})$ and ranged from 0.307 to 0.972 with an average of 0.688±0.141 (n=97). A strong seasonal pattern was observed for both $[NH_3]$ and $fNH_3$, with the highest values observed during warmer periods. No significant

seasonal pattern was observed for $[pNH_4^+]$ that remained relatively consistent throughout each season and characterized by frequent spike events in cold and warm months, including near July 4th, corresponding to a period of significant firework activity.

The $[NH_3]$ and $fNH_3$ were positively correlated with temperature (r = 0.66; p<0.01 & r = 0.51; p<0.01; Figure S1). This

relationship was consistent with previous observations in rural and urban locations that suggested $[NH_3]$ to be influenced by temperature-dependent volatilization (e.g., agriculture, vegetation, sewage, and waste) and evaporation from semi-volatile $NH_4NO_3$ particles (Wang et al., 2015; Hu et al., 2014; Yao et al., 2013; Nowak et al., 2006; Yao and Zhang, 2016). Additionally, $[NH_3]$ was negatively correlated with wind speed (r = -0.42; p < 0.01) and mixing height (r = -0.52; p<0.01) indicating the importance of dilution and vertical height to near-surface $[NH_3]$. The measured $[pNH_4^+]$ were not significantly

correlated with any meteorological parameter (Figure S1). Instead, the annual and seasonal $[pNH_4^+]$ was closely associated with $[pNO_3^-]$ (r=0.69; p<0.01) and $[pSO_4^{2-}]$ (r=0.63; p<0.01). This finding is expected due to the role that $NH_3$ has in neutralizing atmospheric nitric acid and sulfuric acid, leading to $pNH_4^+$ aerosols in the form of $NH_4NO_3$, $NH_4HSO_4$, and $NH_4SO_4$.



### 3.2 Comparison of Urban NH₃ and pNH₄⁺ to Regional Observations

The measured urban [$NH_3$] and [$pNH_4^+$] data from Providence, RI, US was compared with the nearby regional observations from AMoN/CASTNET sites within New England (Figure 1 & Figure 3). Overall, the annual average [$NH_3$] in Providence, RI, was significantly greater ($p<0.05$) than the regional New England AMoN sites; however, [$NH_3$] grouped by season indicates subtle differences in the seasonal profiles at the varying New England sites (Figure 3A). [$NH_3$] at Providence, RI was statistically higher ($p<0.05$) during winter and autumn than the New England AMoN sites and higher than all sites except for Abington, CT, during summer. During spring, [$NH_3$] at Providence, RI, was not statistically different from any of the New England AMoN sites, which typically exhibited a springtime [$NH_3$] peak that likely reflects the influence and timing of fertilization application (Felix et al., 2017). We note that there can be large heterogeneity in urban [$NH_3$]; however, the Providence, RI monitoring site was specifically chosen since it was away from any direct emission sources and at a raised elevation. The difference in our measured [$NH_3$] and reported by AMoN are unlikely to be explained by differences in sampling methodology. We have recently demonstrated that our active denuder sampling technique resulted in $NH_3$ concentrations within 2-5% of that determined from simultaneous deployed passive $NH_3$ collection techniques, which are utilized at AMoN sites (Walters et al., 2020). This result was consistent with previous comparisons between active and passive $NH_3$ sampling techniques (Zhou et al., 2019; Puchalski et al., 2015).

The annual average [$pNH_4^+$] at the Providence, RI site was also found to be significantly higher than the regional CASTNET sites ($p<0.05$; Fig. 3B). However, when broken down by season, the Providence, RI site has significantly higher [$pNH_4^+$] than all the regional CASTNET sites only during autumn ($p <0.05$), suggesting that [$pNH_4^+$] may be more regional representative than [$NH_3$] due to its extended atmospheric lifetime relative to $NH_3$ (Paulot et al., 2016). During the winter and summer, the Providence, RI site did not have significantly higher [$pNH_4^+$] than any of the CASTNET sites. During the spring, [$pNH_4^+$] was higher in Providence, RI, than the two most remote regional CASTNET sites, including Ashland, ME, and Woodstock, NH ($p<0.05$), but not significantly different from the Abington, CT or Underhill, VT sites. It is important to note that methodology differences in the collection of $pNH_4^+$ could have significantly influenced the [$pNH_4^+$] annual differences and seasonal patterns. Our collection method (Nylon filter + acid-coated filter) should lead to the quantitative collection of $pNH_4^+$ (Walters et al., 2019; Yu et al., 2006). In contrast, $pNH_4^+$ collections at the CASTNET sites utilize PTFE filters which could be biased low due to the potential for significant loss of semi-volatile $NH_4NO_3$ (Ashbaugh and Eldred, 2004; Yu et al., 2005). The potential for $NH_4NO_3$ volatilization should be more significant for warmer temperatures (Ashbaugh and Eldred, 2004; Yu et al., 2005). However, we did not observe a significant difference in summer [$pNH_4^+$] between the Providence, RI, and regional CASTNET sites. Thus, the influence of sampling methodologies on the spatiotemporal [$pNH_4^+$] patterns remains difficult to quantify.



Localized $NH_3$ emissions likely play an important role in contributing to the observed elevated urban $[NH_x]$ and the spatiotemporal patterns across New England (Figure 4). The NEI-14 emission profiles at the AMoN sites indicated that agricultural activities drive the seasonal $NH_3$ emissions, while non-agricultural sources, including stationary fuel combustion
(electricity generating units and residential heating) and vehicles, were important during winter but their relative contributions significantly decreased during warmer periods. In contrast, the annual $NH_3$ emission in Providence, RI were dominated by fuel combustion emissions. The total $NH_3$ emission density in Providence, RI had less seasonal variability than the regional AMoN/CASTNET locations despite a potential seasonal change in emissions with relatively high contributions from residential heating (i.e., oil, gas, wood combustion) during winter compared with summer. We note that natural gas and oil
stationary fuel combustion, which is predicted to be the main $NH_3$ emission source at our urban study site as well as in other major urban areas in regions with a large heating demand (Zhou et al., 2019), has a highly uncertain $NH_3$ emission factor established from limited studies conducted before 1982 (Muzio and Arand, 1976; Cass et al., 1982). Additionally, it has been recently pointed out that vehicle $NH_3$ emission, another major source of urban $NH_3$, might be underpredicted by at least a factor of 2 in the NEI (Sun et al., 2017; Fenn et al., 2018).

### 3.3 Urban $\delta^{15}N$ of Urban $NH_x$

Measurements of $\delta^{15}N$ at the Providence, RI monitoring site were utilized to enhance understanding of source contributions to urban $NH_x$. The measured $\delta^{15}N(NH_3)$ ranged from -21.4 to -2.0‰ with an average of -11.9±5.0‰ (n=90), and $\delta^{15}N(pNH_4^+)$ ranged from -7.4 to 17.5‰ with a mean of 4.9±6.2‰ (n=60) (Figure 5). The measured $\delta^{15}N$ data was binned by season that
included winter (Dec, Jan, Feb), spring (Mar, Apr, May), summer (Jun, Jul, Aug), and autumn (Sep, Oct, Nov). The $\delta^{15}N(NH_3)$ was statistically higher during spring (-7.6±3.5‰, n=21 ($\bar{x}±1\sigma$)) compared to the other seasons (summer = -13.9±4.1‰, n=21; autumn = -13.1±5.1‰, n=21; winter = -13.4±5.2‰, n=18, $p<0.05$). The $\delta^{15}N(pNH_4^+)$ also indicated significant seasonality with lower values during summer (0.4±4.9‰, n=18) compared to autumn (7.4±4.8‰, n=15) and winter (9.0±5.8‰; n=14) ($p<0.05$). However, springtime $\delta^{15}N(pNH_4^+)$ (4.1±5.2‰, n=13) was not statistically different from any season.

The $\delta^{15}N$ of atmospheric $NH_3$ and $pNH_4^+$ reflects a combination of source effects from different $NH_3$ emission sources and isotopic equilibrium between $NH_3$ and $pNH_4^+$ that has been shown to have a large influence on setting the N isotopic distribution between these molecules (Walters et al., 2018; Savard et al., 2017; Kawashima and Ono, 2019). Indeed, the annual $\delta^{15}N(pNH_4^+)$ was statistically higher than $\delta^{15}N(NH_3)$ ($p<0.01$), reflecting the contributions from the nitrogen isotope exchange
reactions between $NH_3$ and $NH_4^+$ (R2-R4), which tends to elevate the $\delta^{15}N(pNH_4^+)$ relative to $\delta^{15}N(NH_3)$ (Walters et al., 2018; Kawashima and Ono, 2019; Urey, 1947). The isotope difference or isotope enrichment factor ($^{15}\varepsilon_{pNH4+/NH3}$) between $\delta^{15}N(pNH_4^+)$ and $\delta^{15}N(NH_3)$ was calculated as the following (Eq. 3):

$$\Delta\delta^{15}N \approx {}^{15}\varepsilon_{pNH_4^+/NH_3} = \delta^{15}N(pNH_4^+) - \delta^{15}N(NH_3) \qquad \text{(Eq. 3)}$$



The $\Delta\delta^{15}N$ ranged from -0.1 to 34.1‰ and averaged 17.6±7.8% (n=56) (Figure S2). The wide range of $\Delta\delta^{15}N$ values was generally within the varying $NH_x$ isotope equilibrium reactions (R2-R4) (Walters et al., 2018). However, kinetic isotope fractionation and fresh $NH_3$ emissions that might perturb the $NH_3/pNH_4^+$ isotope equilibrium may also play a role (Pan et al., 2016). The $\Delta\delta^{15}N$ were weakly correlated with temperature (r = -0.55, p <0.01) and $[pNO_3^-]$ (r = 0.33, p<0.05), suggesting that these values were difficult to predict. This result has important implications for previous $\delta^{15}N$ source apportionment studies of $NH_3$ and $pNH_4^+$, which commonly utilize an assumed and theoretically calculated phase-dependent fractionation.

To account for the complex phase-dependence on $\delta^{15}N$ variabilities, we calculated $\delta^{15}N(NH_x)$ according to the following (Eq. 4):

$$\delta^{15}N(NH_x) = fNH_3 \times \delta^{15}N(NH_3) + (1 - fNH_3) \times \delta^{15}N(NH_4^+) \qquad \text{(Eq. 4)}$$

The annual $\delta^{15}N(NH_x)$ ranged from -17.4 to 6.3‰ and averaged -6.0±4.9‰ (n=56) (Figure 5). There was significant seasonality with lower values during summer (-9.0±4.2‰, n=18) compared to winter (-3.4±5.3‰, n=13) and spring (-3.8±3.3‰, n=10). The autumn $\delta^{15}N(NH_x)$ (-6.2±4.1‰, n=15) was not significantly different from any season. The $\delta^{15}N(NH_x)$ is independent of the phase $\delta^{15}N$ fractionation, such that it should be a robust tracer reflecting the integrated source contributions from locally emitted and transported $NH_3$ and $pNH_4^+$. Therefore, the $\delta^{15}N(NH_x)$ observations would suggest a seasonal change in sources of $NH_x$ with increased relative emissions from a source with a high $\delta^{15}N(NH_3)$ value during the colder periods of winter and spring and a lower $\delta^{15}N(NH_3)$ value during summer. Vehicle emissions have an elevated $\delta^{15}N(NH_3)$ value of 6.6±2.1‰ (Walters et al., 2020; Song et al., 2021), such that the relative importance of vehicle emissions to $NH_x$ in Providence, RI may have increased during colder seasons. The observed $\delta^{15}N(NH_x)$ decrease during summer and increase in $[NH_3]$ might suggest increased emissions from temperature-dependent emission sources with a relatively low $\delta^{15}N(NH_3)$ signature, such as volatilization (Felix et al., 2013; Freyer, 1978; Heaton, 1987; Chang et al., 2016; Hristov et al., 2009). There were often large $\delta^{15}N(NH_x)$ variations within each season, which may be related to wind direction shifts and varying contributions from local urban $NH_3$ emission sources and long-range transport of $NH_x$.

### 3.4 Identifying Urban Local Sources of $NH_x$

Wind data and bivariate plot statistical analysis were utilized to investigate local and transported sources of urban $NH_x$. The local wind data indicated a clear shift in wind direction and speed from generally faster winds from the west/northwest during winter to slower winds from the south/southeast and northeast during summer (Figure 6). Wind direction and wind speed polar bivariate CBPF plots of $[NH_3]$ and $[pNH_4^+]$ indicated relative high probability under conditions of low wind speeds (i.e., < 2 m/s) for all seasons, suggesting the importance of local emitted $NH_3$ sources and $pNH_4^+$ formation. These elevated CBPF probabilities were also associated with winds from the southeast to west, the direction of I-195 and I-95, major interstate highways, and industrial sources (Figure 1). The highest $\delta^{15}N(NH_x)$ values within each season were observed with winds from



these directions, implicating the importance of vehicle emissions, which have an elevated $\delta^{15}N(NH_3)$ signature of 6.6±2.1‰ compared to other $NH_3$ sources that tend to have $\delta^{15}N(NH_3)$ values below 0‰, including available industrial $\delta^{15}N(NH_3)$ emissions (Walters et al., 2020).

Additionally, high CPF probabilities for both [$NH_3$] and [$pNH_4^+$] were observed during the warmer seasons of summer and autumn from moderate winds (2-4 m/s) from the northeast and west. This result may implicate local temperature-dependent $NH_3$ emission sources such as sewerage lines, trash cans, soil emissions from green spaces, and regional transport.    These winds were associated with a relatively low $\delta^{15}N(NH_x)$, consistent with volatilization contributions with a low $\delta^{15}N(NH_3)$ emission signature between -56.1 to -10.3‰ based on livestock waste and fertilizer studies (Heaton, 1987; Freyer, 1978; Felix
et al., 2013; Chang et al., 2016). Low CPF probabilities for both [$NH_3$] and [$pNH_4^+$] were generally associated with high wind speeds (i.e., > 4 m/s), reflecting the dilution of these pollutants and strong background mixing. An exception to this trend was observed for [$pNH_4^+$] during the winter, which had elevated CPF probabilities with high wind speeds indicating the importance of long-range transport. Interestingly, there was a seasonal difference in $\delta^{15}N(NH_x)$ from this wind profile, with high values during the cold seasons and low values during summer, suggesting that the background $NH_x$ had larger contributions from
vehicle emissions and volatilization during the cold and warm seasons, respectively.

### 3.5  Role of Long-Range Transport as a Source of Urban $NH_x$

Air mass back trajectories and PSCF analysis were utilized to identify source locations of transported $NH_3$ and $pNH_4^+$ to Providence, RI. The clustered seasonal air mass back trajectories indicated a shift in the seasonal air mass origin with winds
originating from the north and west during winter with higher contributions of air masses derived from the south and along the coast during summer (Figure 7). During summer and autumn, potentially significant $NH_3$ and $pNH_4^+$ source regions originated over the mid-Atlantic, midwestern US, Atlantic coast, southeastern US, southeastern Ontario, and southeastern Quebec. These regions have significant agricultural-related $NH_3$ emissions such as fertilizer application and livestock waste. Transport from these regions tended to have relatively low mean $\delta^{15}N(NH_x)$ values (i.e., -15 to -5‰), consistent with transport of volatilized
agricultural $NH_3$ emissions that favor the release of isotopically light $^{14}NH_3$ (Heaton, 1987; Freyer, 1978; Felix et al., 2013; Chang et al., 2016). Indeed, available ground-based monitoring data indicates these regions tend to have elevated ambient [$NH_3$] and [$pNH_4^+$], consistent with these regions as potential $NH_3$ and $pNH_4^+$ source contributors to Providence, RI (Figure S3). The high CBPF probability from southeast Canada may also represent wildfire $NH_3$ emissions, as summer and autumn typically correspond to high wildfire activity in this region (Matz et al., 2020). Direct wildfire $\delta^{15}N(NH_3)$ emission signatures
are unknown; however, emissions from biomass burning (coal combustion) report an emission signature of -6.1±1.3‰ (Freyer, 1978), which is within the range of values observed from the potentially contributing biomass burning regions. Additionally, the Atlantic coast may represent contributions from ocean $NH_3$ flux, which has been suggested to have low $\delta^{15}N$ values (Jickells et al., 2003).





Elevated PSCF probabilities were identified for [pNH$_4^+$] during the winter from the mid-Atlantic and Midwestern US, which is consistent with available [pNH$_4^+$] ground-based observations that tend to peak during this period due to ambient conditions that favor the formation of NH$_4$NO$_3$ (Figure S3). This transport region tended to have relatively high mean $\delta^{15}$N(NH$_x$) values (e.g., -5 to 0‰) from the Midwestern US and relatively low mean from the Mid-Atlantic (~-10‰). Across the US, [NH$_3$] was lowest during winter due to decreased agricultural activities (Figure S3). Indeed, the NEI-14 indicates that the relative

importance of non-agricultural NH$_3$ sources increases during winter (Figure 4), such that the higher $\delta^{15}$N(NH$_x$) values deriving from the Midwest may reflect the regional importance of sources with an elevated $\delta^{15}$N(NH$_3$) value such as vehicles and/or fuel-combustion. Lower mean $\delta^{15}$N(NH$_x$) values derived from the mid-Atlantic may suggest that agricultural emissions such as animal housing remain an important wintertime NH$_x$ source contributor to Providence, RI. Additionally, there could be contributions from fuel combustion with selective catalytic reduction technology that have a reported $\delta^{15}$N(NH$_3$) signature of

-14.6 to -11.3‰ (Felix et al., 2013).

### 3.5 Urban NH$_x$ Source Apportionment

The NH$_x$ source contributions at Providence, RI, including local and transported emissions, were quantified using SIMMR (Parnell et al., 2010). The model was initiated using the measured $\delta^{15}$N(NH$_x$) values and assuming vehicles, volatilization,

fuel combustion with selective catalytic converters (SCR), industry, and biomass burning were the main sources, as evidenced by the local wind direction and back trajectory analysis and the NEI-14 predictions. We acknowledge that there are additional miscellaneous NH$_3$ sources in an urban environment, including pets, household products, and humans; however, we assumed that these sources were negligible compared to the main identified emission sources.

The source apportionment results are sensitive to the number of considered sources, their designated $\delta^{15}$N(NH$_3$) emission signatures, and uncertainty. The input $\delta^{15}$N(NH$_3$) emission source signatures were deliberately chosen from sampling methodologies that have utilized active sampling approaches, as it has been well-documented from several studies that passive samplers result in a $\delta^{15}$N(NH$_3$) bias and could be unreliable (Pan et al., 2020; Kawashima et al., 2021; Walters et al., 2020). Fertilization application is a significant source of NH$_3$ emissions globally and within the US. However, fertilizer application

represents a small component of the overall agricultural emissions at our site (~1.8%) and within our region (7.1%; US EPA Region 1) based on the NEI-14. Further, fertilization-related NH$_3$ emissions tend to peak during spring; however, we neither identified any significant NH$_x$ long-range transport region nor observed a relative decrease in $\delta^{15}$N(NH$_x$) during spring, which would be consistent with a suspected low fertilizer volatilization $\delta^{15}$N(NH$_3$) emission signature. Thus, fertilizer application was not considered in our source apportionment model.






The input source values for vehicles, fuel combustion (SCR)/industry, biomass burning/coal, and volatilization were fixed at 6.6±2.1‰ (Walters et al., 2020), -15.3±3.6‰ (Heaton, 1987; Freyer, 1978), -6.1±1.3‰ (Freyer, 1978), and -19.2±8.3‰ (Freyer, 1978; Heaton, 1987; Hristov et al., 2009; Frank et al., 2004). Fuel combustion with SCR and industry $\delta^{15}N(NH_3)$ emission signature was grouped due to similar values. The volatilization $\delta^{15}N(NH_3)$ emission signature represents integrated

volatilization measurements conducted in animal sheds (Freyer, 1978; Heaton, 1987), and measurements that include monitoring volatilization as a function of time, which indicate significant $\delta^{15}N(NH_3)$ variability (Hristov et al., 2009; Frank et al., 2004). Further details on our rationale for the chosen source $\delta^{15}N$ values are provided in the Supporting Information (Text S1).

The mixing model predicts that vehicles, volatilization, fuel-combustion (SCR)/industry, and biomass/coal contribute (mean±σ) 30.7±11.6%, 17.0±10.8%, 18.3±12.1%, and 33.9±23.4% to the annual $NH_x$ background in Providence, RI (Figure 8). The relative contribution of vehicle emissions had a strong seasonal profile with higher contributions during the colder seasons of spring (42.9±11.3%) and winter (45.8±13.0%) compared to the warmer seasons of summer (20.8±9.7%) and autumn (29.5±12.2%). The relative contribution for volatilization and fuel combustion (SCR)/industry was predicted to peak during

summer with means of 24.9±13.8% and 24.8±15.9%, compared to winter with means of 14.1±8.7%, 15.1±10.0%, respectively. Biomass/coal emissions were relatively consistent across seasons, with a peak during autumn of 33.3±22.6%.

The input emission signatures for volatilization and fuel combustion (SCR)/industry have somewhat overlapping values, such that it was difficult for the mixing model to differentiate between these sources. Based on the NEI-14, wind direction, and

long-range transport analysis (Figures 4, 6, & 7), we suspect the relative contribution of vehicle emissions diminished during summer due to the importance of temperature-dependent $NH_3$ volatilization emissions. The exact $NH_3$ volatilization source remains unclear. There was evidence of significant contributions from local urban volatilization (i.e., sewage, waste, urban green spaces) and long-range transport from regional agricultural activities and ocean flux (Figures 6 and 7). Stationary fuel combustion and industry $NH_3$ emissions were unlikely to explain the seasonal shift in the relative increase of non-vehicle

emissions observed during summer, as the NEI-14 predicts stationary fuel combustion emissions as more important contributors during winter due to significant heating demand, and industrial emissions were expected to have a non-seasonally dependent emission profile (Figure 4). Interestingly, a relatively low fractional contribution was predicted for fuel combustion (SCR) for winter, despite the NEI-14 indicating that residential fuel (natural gas and oil) combustion was the largest emission source of $NH_3$ at our study site and other cities with significant heating demands (Zhou et al., 2019). While we acknowledge

that the fuel combustion $\delta^{15}N(NH_3)$ emission signatures were uncertain, the mixing model and seasonal [$NH_3$] results would suggest that residential $NH_3$ emissions were overpredicted in the NEI. Their emission factors may need to be revisited to more accurately model urban [$NH_3$] and predict its human and ecological impacts. Biomass burning could be a seasonal consistent contributing $NH_x$ source due to the role of residential wood combustion (Figure 4) during colder months and wildfires during the warmer months. However, the $\delta^{15}N$ source apportionment results for biomass burning predict an extremely broad



distribution (Figure 8; Figure S4). Future $NH_x$ source apportionment of biomass burning should consider additional biomass tracers such as potassium and levoglucosan.

## 4. Conclusion

Elevated urban $NH_x$ concentrations were observed in Providence, RI, relative to regional background monitoring stations in New England. Mixing model $\delta^{15}N(NH_x)$ source apportionment results utilizing $\delta^{15}N(NH_x)$, suggest that vehicles represent an
important source of urban $NH_x$ with strong seasonal variability. The relative contribution of vehicle emissions was highest during winter/spring, which is significant because $NH_3$ emissions may contribute to the elevated $PM_{2.5}$ observed during this time in the eastern US (Shah et al., 2018). Reductions in vehicle ammonia emissions may represent a promising way to mitigate the adverse impacts of elevated urban $NH_3$ concentrations and yield positive benefits for ecosystems and human health. However, vehicle $NH_3$ emissions are a consequence of the technology used to combat vehicle $NO_x$ and CO emissions.
Decreasing vehicle $NH_3$ emissions may not be achievable until the vehicle fleet electrification. Expanding national observational networks to include urban measurements of $[NH_3]$ and $\delta^{15}N(NH_x)$ are needed to monitor urban trends and design future regulatory $NH_3$ fossil-fuel-related emission reductions.

This work demonstrated that nitrogen isotopic analysis allows for further refinement of our understanding and quantification
of urban $NH_x$ sources, laying the foundation for future source apportionment studies. Utilizing a laboratory-verified collection method suitable for $NH_x$ speciation and isotope analysis was critical for accurate source apportionment due to the observed complex phase-dependent $\delta^{15}N$ isotope fractionation between $NH_3$ and $pNH_4^+$. Future studies should improve our understanding of the drivers behind $NH_3$ and $pNH_4^+$ phase $\delta^{15}N$ fractionation, including controlled chamber studies and field observations, which may also provide important insights into controls on $NH_3/pNH_4^+$ gas to particle-phase conversion. Still,
this work highlights the need to improve our $\delta^{15}N(NH_3)$ emission source values, particularly for our volatilization, industry, fuel combustion, and biomass burning sources, to enhance the quality of the source apportionment results.

**Data Availability.** Data presented in this article are available on the Harvard Dataeverse at
https://doi.org/10.7910/DVN/JHMBRI and in the Supplement .

**Author contributions.** WWW, MK, and MGH designed varying aspects of the field sampling plan. WWW, MK, and DEB carried out the field measurements. WWW, MK, and DEB conducted all laboratory analyses of data. EW contributed spatial analysis of data. BHB contributed emission modelling of the presented data. WWW prepared the article with contributions
from all co-authors.





**Competing interests.** The authors declare that they have no conflict of interest.

**Acknowledgements.** We thank Ruby Ho for sampling and laboratory assistance. We are grateful to Paul Theroux of RI-
DEM/RI-DOH for access and support at the RI-DEM air-monitoring sites.

**Financial support.** This research has been supported by the National Science Foundation Division of Atmospheric and
Geospace Sciences (grant: 1624618) and the Institute at Brown for Environment and Society (internal grant no. GR300123).

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



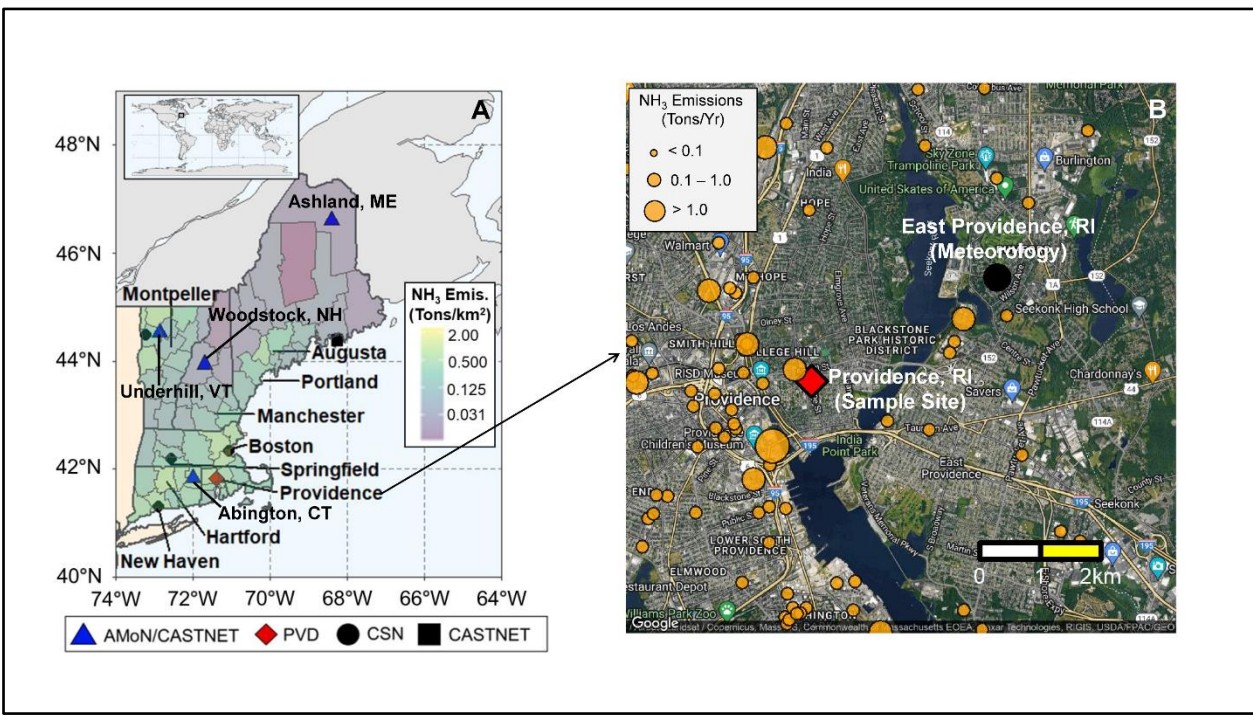

**Figure 1.** **Overview of the sampling location in Providence, RI, USA (red diamond) located within New England (A) with the Ammonia Monitoring Network (AMoN)/Clean Air Status and Trends Network (CASTNET; blue triangle), Chemical Speciation Network (CSN; black circle), CASTNET only (black square) monitoring locations indicated. The counties in A are color-coded for NEI-14 $NH_3$ emission densities. The zoomed-in map of Providence, RI, US is shown in B with the sample site location (red diamond), the nearby CSN location with reported meteorology data (black circle) in East Providence, RI, USA, and the $NH_3$ point emission sources from the NEI (orange circles; size-coded to annual $NH_3$ emission) indicated. Image (B) was created using © Google Maps (Map data ©2019 Google).**





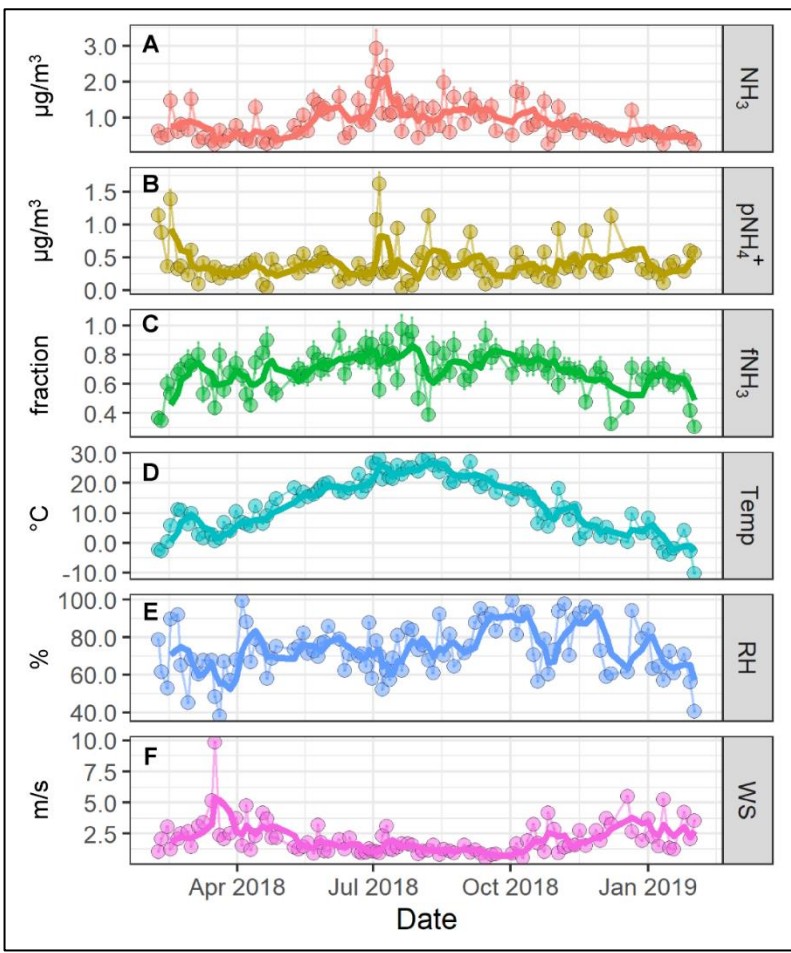

**Figure 2. Time series plots of the measured NH$_x$ data including (A) [NH$_3$], (B) [pNH$_4$$^+$], and (C) fNH$_3$ and the reported meteorology data including (C) temperature (Temp), relative humidity (RH), and wind speed (WS) from Feb 2018 – Feb 2019 in Providence, RI, US. The light data points refer to the 24-h integrated samples (A, B, C) or 24-h averaged meteorology data (D, E, F), and the dark lines represent approximate 2-week moving averages.**





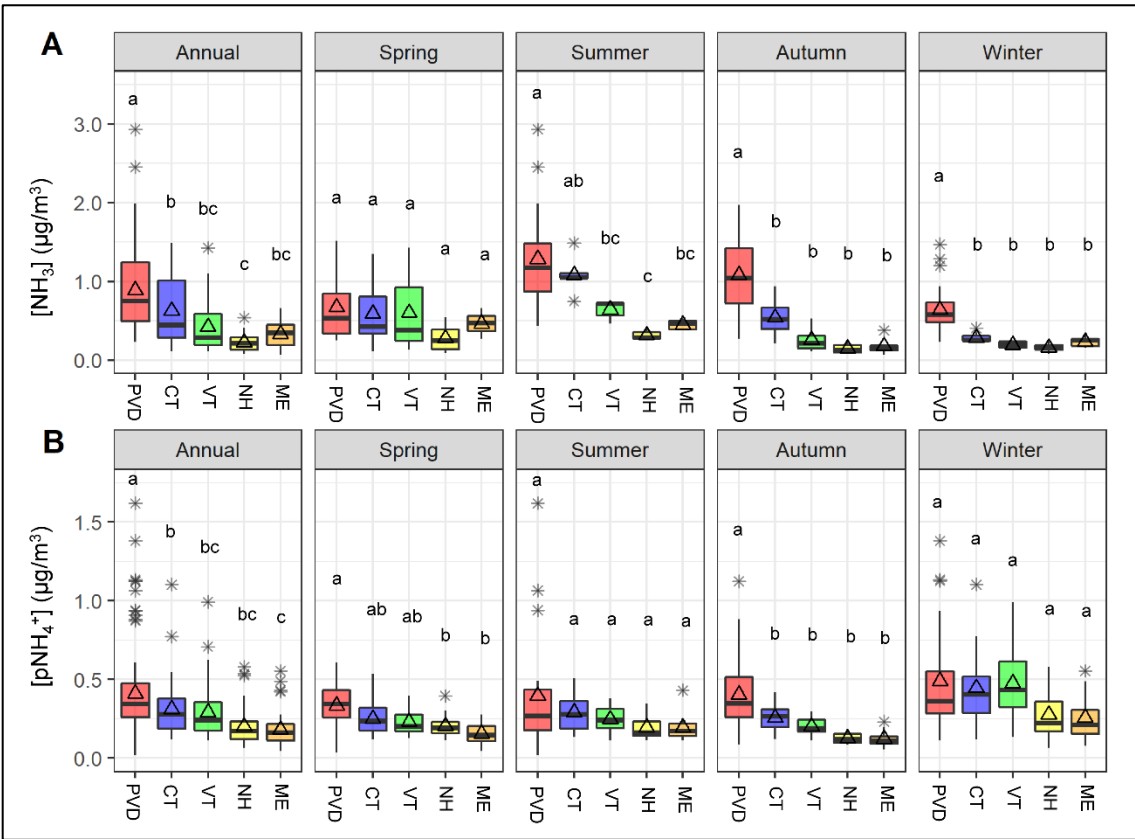

**Figure 3.** Box and whiskers plots that summarize the annual and seasonal (A) [NH₃] and (B) [pNH₄⁺] distributions (lower extreme, lower quartile, median, upper quartile, and upper extreme) with the mean (open triangle) and outlier (black asterisk) at the Providence, RI (PVD) site and the New England AMoN/CASTNET sites including Abington, CT (CT), Underhill, VT (VT), Woodstock, NH (NH), and Ashland, ME (ME). Similar lowercase letters in the box and whiskers plots represent categories with statistically similar values.





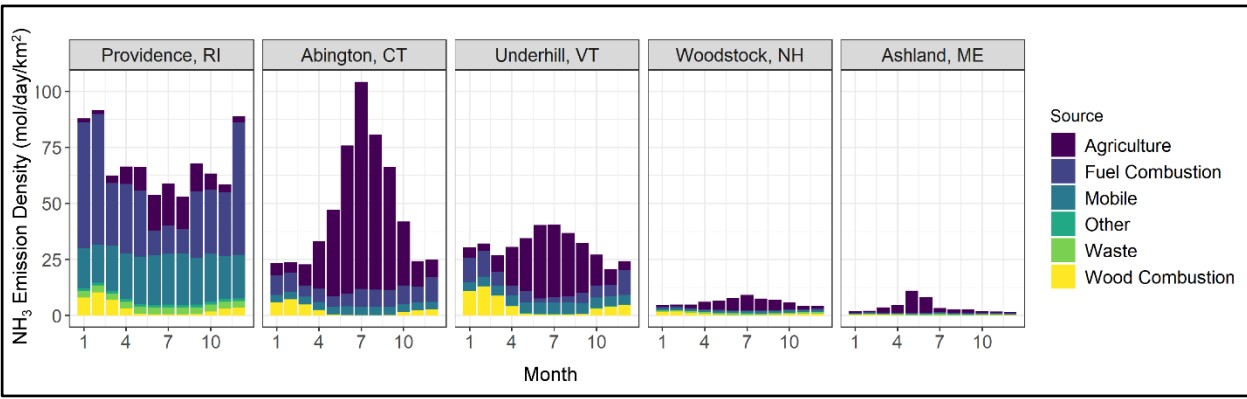

**Figure 4.** Monthly-based NH₃ emission densities speciated between agricultural, fuel combustion, mobile, other, waste, and wood combustion computed by the SMOKE model for the counties of the New England NH₃ monitoring site.





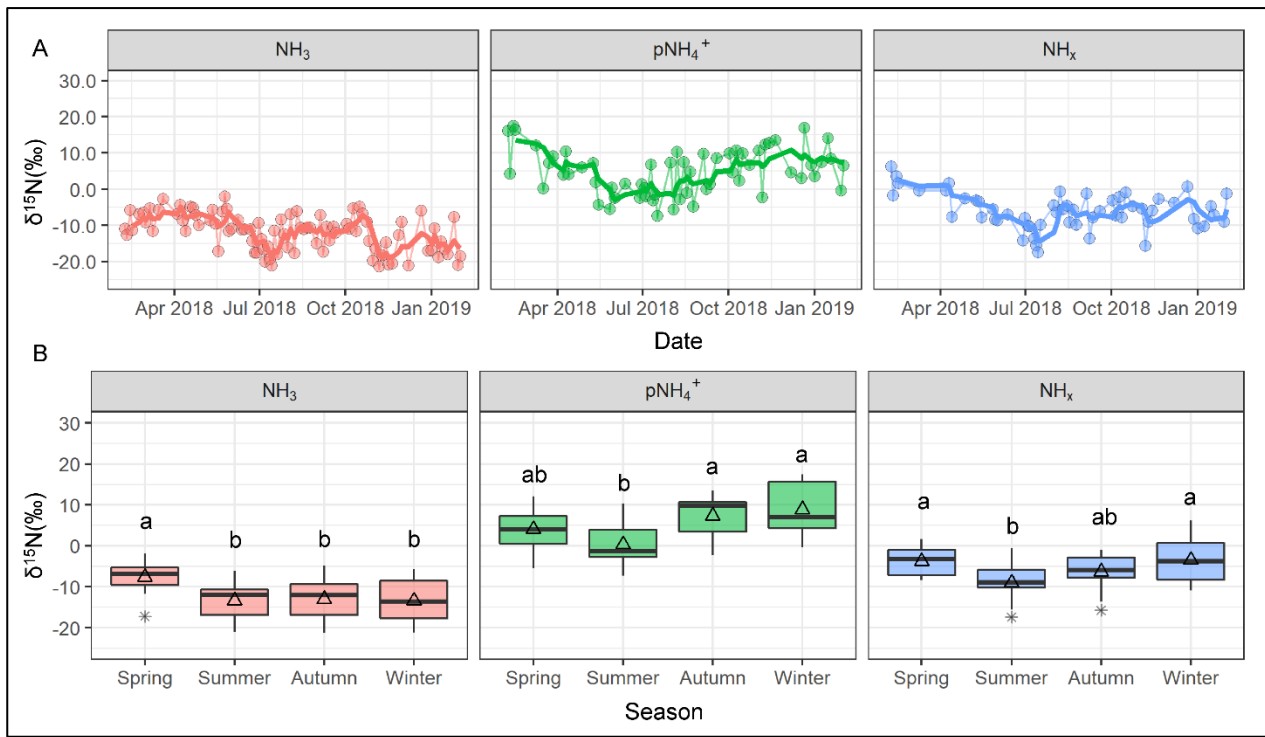


**Figure 5. Measured $\delta^{15}$N data of NH₃, pNH₄⁺, and NH$_x$ collected in Providence, RI, including (A) time series and (B) seasonal box and whiskers plots summarizing the distributions (lower extreme, lower quartile, median, upper quartile, and upper extreme) with the mean (open triangle) and outlier (black asterisk). Similar lowercase letters in the box and whiskers plots represent categories**
**with statistically similar values.**







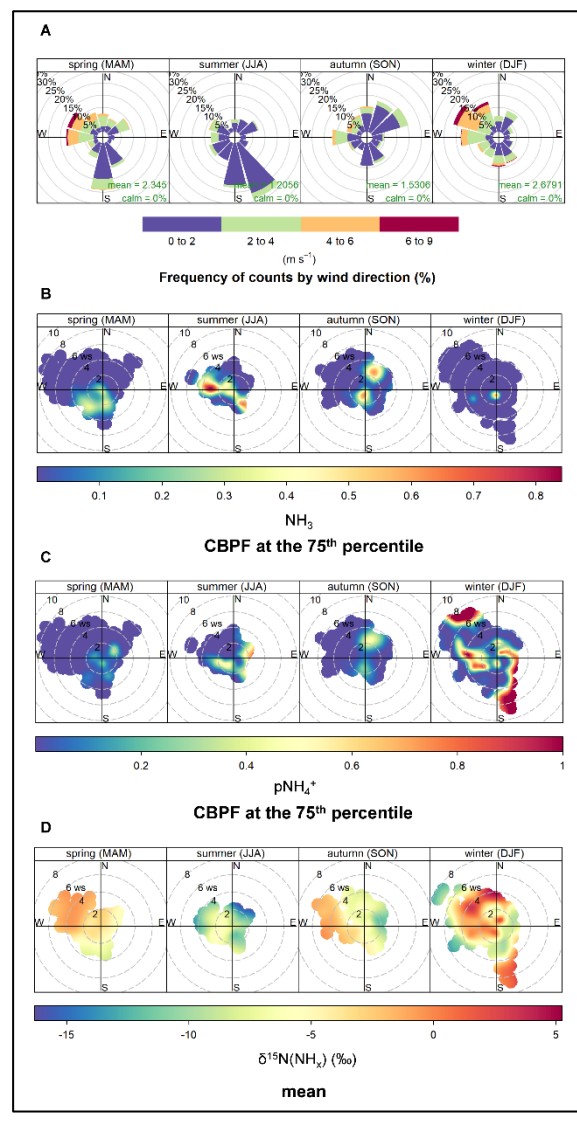

**Figure 6. Overview of (A) windrose plots and polar bivariate (wind direction and wind speed) plots of the conditional bivariate probability function (CBPF) for (B) [NH₃] and (C) [pNH₄⁺], and (D) mean δ¹⁵N(NHₓ) in Providence, RI, sorted by season.**




**Figure 7. Influence of long-range transport including (A) clustered seasonal air mass back trajectories, (B) seasonal [NH₃] potential source contribution function probability (PSCF), (C) seasonal [NH₃] PSCF probability, and (D) seasonal air mass back trajectory δ¹⁵N(NHₓ) mean values.**





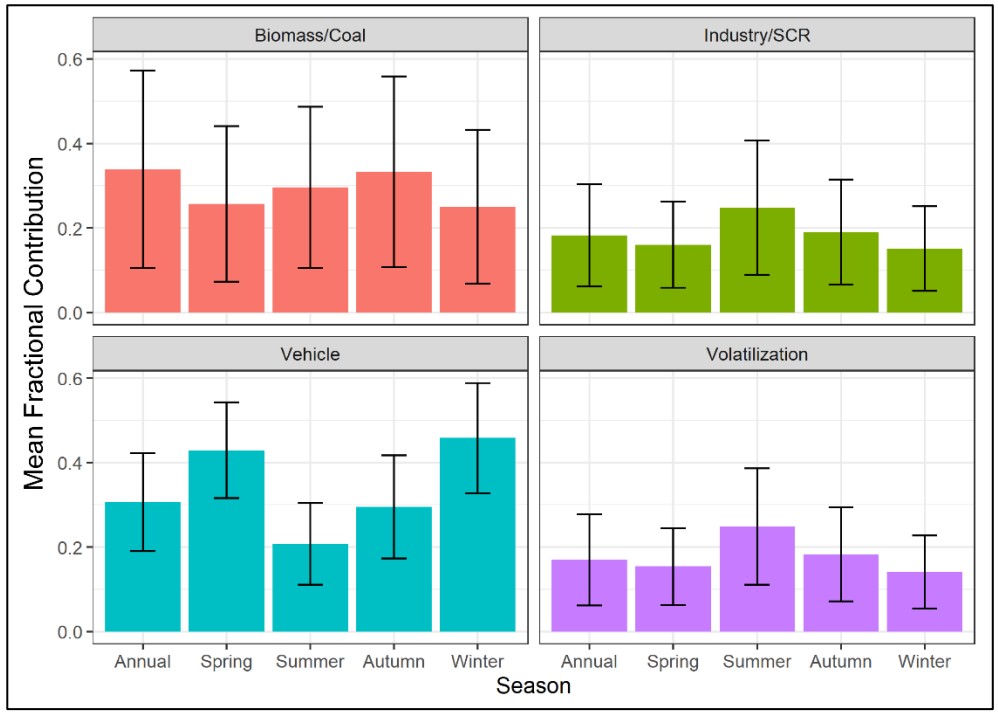

**Figure 8.** The calculated mean seasonal and annual fractional contribution of emission sources (Biomass/Coal, Industry/SCR, Vehicle, Volatilization) to $NH_x$ in Providence, RI, utilizing a stable isotope mixing model (SIMMR). The error bars represent the standard deviation of the model simulations.
