# Peer review of "Quantifying the Importance of Vehicle Ammonia Emissions in an Urban Area of the Northeastern US Utilizing Nitrogen Isotopes"

_Atmospheric Chemistry and Physics, 2022_

## Author Comment (AC1)

We appreciate both Reviewers' helpful and constructive feedback, which has helped improve our manuscript overall. Specifically, we have improved our discussion of long-range transport, as suggested by Reviewer #1, to focus on the potential for significant source contributions from heavily urbanized Toronto and the eastern shoreline while drawing focus away from biomass burning. These changes have been implemented throughout our revised manuscript. We have also updated our mixing model to exclude the role of biomass burning since there was no significant evidence that it was a major contributing source to our study region. These changes do not alter the major findings of this work that vehicle emissions are an important source of urban $NH_3$. We have expanded our discussion of isotope fractionation effects observed between $NH_3$ and $pNH_4^+$, improved our source apportionment discussion, and compared our results with the National Emission Inventory as recommended by Reviewer #2. Overall, these changes have led to improvement of the presented manuscript. A point-by-point response to all reviewer comments is provided below.

**Reviewer #1:**

**Overview:** The Authors present a detailed foray into stable isotope source apportionment of NHx in a urban area to ascertain the relative importance in accounting for vehicle and industry NH3 emissions when setting air quality policy. They combine a robust observational dataset and isotope mixing model with archived NHx observations to unravel the sources contributing to ambient levels in Providence, RI. Overall, the manuscript presents a strong case for the prevalence and significant contribution of vehicular NH3 to urban air. Pending minor revisions below, the manuscript will be suitable for publication in Atmospheric Chemistry and Physics.

**Response:** Thank you for your helpful and constructive feedback. We have addressed the raised points and revised the manuscript according to these suggestions.

**Comment 1:** The most substantial oversight in interpreting the long-range transport component of the NHx observations comes from the inferred agricultural regions of Canada that are upwind. While these areas are agriculturally intensive, the Authors seem to have overlooked that one of the 3 largest cities in North America, Toronto, is in this same fetch. It is a potent source of vehicular and industrial NH3 emissions. Similarly, Montreal and additional industry along the St. Lawrence River in Quebec are overlooked. The Authors should consider these and probably revise their stress on wildfire (biomass burning) emissions towards those from more urban and industrial sources.

**Response 1:** We appreciate the feedback and recognize our oversight in the original version of our manuscript when discussing the long-range transport of $NH_x$. Based on the reviewer's suggestions, we have removed all discussions of potential long-range transport of wildfire emissions and removed biomass burning as a potential important $NH_3$ source in our source apportionment model. Thus, we have omitted Page 13, Line 359-362, and Page 14, Lines 422-424, in the originally submitted manuscript in the revised manuscript. We also agree that Canada's

heavily urbanized and industrialized regions could have significantly contributed $NH_x$ to our study site. We have updated Section 3.5 in the revised manuscript to clarify this point, following recommendations based on Comment 7.

**Comment 2:** Page 6, Line 165-168: Given the sampling setup, it seems strange that you'd find NO2- in these extracts. Are the sources for this known? Is it an impurity in the reagents used for coating and sample preparation? A bit of elaboration on this could help reduce confusion on why this issue arises. Maybe give the breakdown of the criteria that were unsatisfied here (e.g. XX% of rejected samples were because of criterio 2)? This way the propensity of each issue is clear.

**Response 2:** Thank you for this suggestion. The presence of $NO_2^-$ was never an issue for either the citric acid-coated denuders or the citric acid-coated cellulose filters. Instead, it was an occasional issue for the Nylon filters. We suspect that $NO_2^-$ may derive from some collection of $NO_2$ that is not completely removed from the upstream carbonate denuder, as previous studies have shown that a fraction of $NO_2$ can be removed by Nylon Filters (Perrino et al., 1988). In the revised manuscript, we have broken down the number of rejected samples based on the three criteria for both $NH_3$ and $pNH_4^+$ samples: "These criteria were met for 90 out of 97 $NH_3$ samples and 60 out of 97 $pNH_4^+$ samples. The 7 rejected $NH_3$ samples were because of criterion 3, while the rejected $NH_4^+$ samples included 18 from criterion 1, 8 from criterion 2, and 11 from criterion 3. The presence of significant amounts of $[NO_2^-]$ was found exclusively on the Nylon filters, which likely reflect the influence of $NO_2$ collection as previously demonstrated (Perrino et al., 1988)". This revision was made on Page 5, Line 129-133 in the revised manuscript. Also, we note that there was a typo in the original version of the manuscript that 89 out of 97 $NH_3$ samples met the criteria when the actual number of samples was 90.

**Comment 3:** Page 9, Line 238: Should 'fertilization' be fertilizer?

**Response 3:** Thank you for catching this typo. We have made this correction on Page 7, Line 205 in the revised manuscript.

**Comment 4:** Page 11, Line 294: Should % instead be ‰?

**Response 4:** Thank you for pointing out this typo, and we have made this correction on Page 9, Line 259 in the revised manuscript.

**Comment 5:** Page 11, Lines 308-310: A technical oversight here that there could also be increased physical losses driving the observations. The dry deposition loss of NH3 to the Earth's surface is sustained during transport, and this would happen more readily in the winter. Suggest revising

with the alternative process also used to provide context. There is plenty of literature on the subject.

**Response 5:** Thank you for pointing out this oversight. We have revised this section also to indicate the potential importance of $NH_3$ physical processing in influencing the $\delta^{15}N(NH_x)$ values, "The physical processing of $NH_3$ could have also played an important role in the observed $\delta^{15}N(NH_x)$ seasonal trends. The enrichment factor associated with $NH_3$ dry deposition has not been measured directly. Still, it has been suggested to be low (~ 4‰) based on the physical processing of $NH_3$ in a vehicle tunnel (Walters et al., 2020). This result would suggest that as $NH_3$ undergoes dry deposition, the pool of $\delta^{15}N(NH_3)$ in the atmosphere becomes slightly depleted as the heavier $^{15}NH_3$ is preferentially deposited. The increased temperatures during summer and autumn would have increased the amount of dry deposited $NH_3$ that re-volatilized into the atmosphere (Behera et al., 2013). $NH_3$ volatilization has been shown to have a significant fractionation effect leading to the emission of $NH_3$ depleted in $^{15}N$ (Hristov et al., 2009; Frank et al., 2004). Further work is needed to refine our understanding of $NH_3$ bidirectional exchange and its impact on $\delta^{15}N$; however, we expect this process would have contributed to lower $\delta^{15}N(NH_x)$ values during the warmer periods due to increased temperature-dependent $NH_3$ volatilization". This correction was made on Page 10, Line 294-303 in the revised manuscript.

**Comment 6:** Page 12, Line 332: 'sewerage' should be 'sewage' and appropriate references need to be added at the end of this sentence.

**Response 6:** Thank you for the comment, and we have made this correction in the revised manuscript. Further, we have added the following references for sewage lines, trash cans, soils emissions from green spaces, and regional transport as sources of urban $NH_3$ at the end of the sentence: Hu et al., 2014; Sutton et al., 2000; Pandolfi et al., 2012; Reche et al., 2012; Meng et al., 2011; Galán Madruga et al., 2018; Zhou et al., 2019. These changes were made on Page 11, Line 320-322 in the revised manuscript.

**Comment 7:** Page 12, Section 3.5: Implement consideration of the minor revision regarding urban and industrial areas in Ontario and Quebec throughout.

**Response 7:** Thank you again for pointing out this oversight. In the revised section 3.5, we have discussed the potential for urbanized regions and industrial areas as sources of long-range transport of $NH_x$ to our study site during the warmer months. In addition to agricultural emissions, we have highlighted the potential for urban and industrial activities on Page 11, Lines 336-338 in the revised manuscript, "These regions have significant agricultural-related $NH_3$ emissions such as fertilizer application and livestock waste, as well as significant urban and industrial activities". Further, we highlighted that the $\delta^{15}N(NH_x)$ deriving from these regions had an overlapping value with available industrial emission measurements on Page 12, Line 340-341 in the revised manuscript, "…and available industrial emissions with reported low $\delta^{15}N(NH_3)$ values of -20.1‰ from a steel factory (Heaton, 1987)". Finally, we acknowledge that physical processing during transport from

these regions of high $NH_3$/$pNH_4^+$ emissions could have also contributed to the observed low $\delta^{15}N$ values on Page 12, Line 341-347 in the revised manuscript, "We also note that $NH_3$ deposition and re-volatilization during transport of any $NH_3$ emission source may also lead to significant isotope fractionation as $NH_3$ was transported downwind. Because $NH_3$ volatilization has been shown to lead to the initial release of $NH_3$ depleted in $^{15}N$, it is reasonable to assume that this long-range transported $NH_3$ would contribute low $\delta^{15}N(NH_3)$ (Frank et al., 2004; Hristov et al., 2009). Thus, low $\delta^{15}N(NH_3)$ values from the identified important contribution regions during the warmer seasons may also reflect the bidirectional exchange of $NH_3$ as it is long-range transported downwind from agricultural, urbanized, and industrialized regions".

**Comment 8:** Page 12, Line 350: Deposition and revolatilization is not limited only to agricultural NH3, but any emitted NH3, which will then hop downwind and fractionate along the way. Suggest revising here.

**Response 8:** Thank you for raising this point. In the original manuscript, we meant to define the $NH_3$ volatilization as all volatilization sources and not just related to agricultural activities. We revised the manuscript to clearly indicate that $NH_3$ volatilization can occur for any emission source as $NH_3$ is transported downwind due to its bidirectional exchange. We have added the following lines in the revised manuscript, "We also note that $NH_3$ deposition and re-volatilization during transport may also lead to significant isotope fractionation as $NH_3$ is transported downwind. Because $NH_3$ volatilization has been shown to lead to the initial release of $NH_3$ depleted in $^{15}N$, it is reasonable to assume that this long-range transported $NH_3$ would contribute low $\delta^{15}N(NH_3)$ (Frank et al., 2004; Hristov et al., 2009). Thus, low $\delta^{15}N(NH_3)$ values from the identified important contribution regions during the warmer seasons may also reflect the bidirectional exchange of $NH_3$ as it is long-range transported downwind from agricultural, urbanized, and industrialized regions". This revision was made on Page 12, Lines 341-347 in the revised manuscript.

**Comment 9:** Page 12, Line 354: Wildfires are not that common. Suggest retuning for urban emissions upwind.

**Response 9:** Thank you for pointing this out. We have removed mention of wildfires in this section and returned focus to the urban emissions upwind. Thus, we have removed Page 13, Line 359-362 in the original manuscript in the revised version.

**Comment 10:** Page 13, Lines 368-370: This is likely mixed in with volatilized NH3 from Canada. Revise the discussion in this section to be a bit more aware of sources in the region.

**Response 10:** Thank you for pointing this out. We have revised the manuscript to indicate that volatilized $NH_3$ emissions from Canada could be important during this period, "…, and contributions from upwind volatilized $NH_3$ emissions from Canada". This correction was made on Page 12, Line 362 of the revised manuscript.

**Comment 11:** Page 13, Line 375: It isn't clear how SCR differs from vehicle sources of NH3? Here and below these are mixed and there doesn't seem to be a definition on why these are considered separate sources? I eventually found a description in the SI, but that example should be given here. From the SI it is not clear whether all power generation in the US is equipped with SCR by law? Couldn't there be quite a lot that is not? If that is the case, should SCR be used so specifically in the main manuscript? Consider revising for clarity.

**Response 11:** Thank you for pointing out that our presentation on differentiating SCR from vehicles was confusing. We have revised the manuscript to differentiate these sources more clearly as vehicles and stationary fuel combustion. Further, we have defined stationary fuel combustion as residential fuel combustion, industrial fuel combustion, and energy generating units. We have removed our mention of SCR (selective catalytic reduction) since not all stationary fuel combustion sources utilize this technology, particularly residential fuel combustion emissions. These changes have been made throughout the manuscript and SI.

**Comment 12:** Page 13, Line 377: Indoor sources requires an appropriate reference(s).

**Response 12:** Thank you for pointing this out. We have added the following references to indoor sources of $NH_3$: Li et al., 2020; Sutton et al. 2000; Ampollini et al., 2019. This correction is made on Page 13, Line 370 of the revised manuscript.

**Comment 13:** Page 14, Lines 409-411: In summer, your source region is strongly influenced by the heavily urbanized Toronto and shoreline area.

**Response 13:** Thank you for pointing this out. We have restructured this section to focus on the change in transport during the summer reflecting contributions from Toronto and shoreline areas that could have contributed to the shift in $NH_3$ emission source apportionment results. These lines were revised as, "Based on the NEI-14, wind direction, and long-range transport analysis (Figures 4, 7, & 8), we suspect the relative contribution of vehicle emissions diminished during summer due to the increased importance of temperature-dependent $NH_3$ volatilization emissions, increased energy consumption due to cooling demands, and/or change in transport over heavily industrialized regions such as heavily urbanized Toronto and the East Coast shoreline. The exact $NH_3$ volatilization source remains unclear. However, there was evidence of significant contributions from local urban volatilization (i.e., sewage, waste, urban green spaces) and long-range transport from regional agricultural regions and over the ocean". This correction was made in the revised manuscript on Page 14, Lines 405-410.

**Comment 14:** Page 14, Line 424: This line of reasoning about wildfires doesn't have much factual justification. Is this speculative or can quantitative metrics be brought to bear on this? Suggest removing throughout if it is speculation as a major Canadian source. Biomass burning/wildfire events intruding into the US are largely confined to those originating in Alberta and British Columbia.

**Response 14:** Thanks again for raising this point. We did not measure any additional tracers of biomass burning in this work, such that we have removed speculation regarding wildfire contributions of $NH_3$ to our study site. In the revised manuscript, we have removed Page 14, Lines 422-424 from the original manuscript.

**Comment 15:** Page 30, Figure 7: Can an accurate representation of the great lakes be added to these figures? Since there's discussion regarding Ontario and Quebec linked to this, the major geographical features that make them identifiable should be properly plotted here.

**Response 15:** That is an excellent suggestion; however, we have used the highest resolution maps available in the "OpenAir" Package and cannot programmatically provide further differentiation between Ontario and Quebec in this figure.

**Comment 16:** Page 4, Figure S3: Given the specific commentary on sources of NH3 from Canada, have the authors looked into acquiring CAPMoN or NADP datasets to include the source region more completely in this visualization?

**Response 16:** This is a nice suggestion, but we did not pull any of the CAPMoN or NADP datasets from Canada in this work, and the contributing author that led this part of the project has since moved on. Since this figure is supplementary to the main analyses, we have decided that there isn't a need for this in the presented article. Still, we will consider this data in future $NH_3$ work for the northeastern US.

**Comment 17:** Page 7: 'may closer represent' should be 'may more closely represent'

**Response 17:** Thank you for catching this typo. We have made this correction accordingly in the revised SI.

**Comment 18:** Page 7: '…. Representative of all industry-related NH3 emissions'. Since steel industry uses coal in the coking process, this isn't too surprising to see a similar trend to the values found from fuel combustion. Values from natural gas or oil-fired facilities would provide a greater

level of confidence that all fossil fuel fired power or industrial activities yield a similar NH3 signature.

**Response 18:** Thank you for pointing this out. We agree having the natural gas, and oil-fired facility $\delta^{15}N(NH_3)$ signatures would be useful; however, to date, this measurement has not been conducted but should be a research topic for the future, along with residential fuel combustion emissions. In the original version of the manuscript, we point out that the fuel combustion $\delta^{15}N(NH_3)$ signatures remain uncertain (Page 14; Lines 419-421). However, since vehicle emissions have a unique $\delta^{15}N(NH_3)$ emission source value, specifically it is the only known source with a positive $\delta^{15}N(NH_3)$ signature, we don't think the highly uncertain fuel combustion signatures will significantly impact the source apportionment results of $NH_3$ deriving from vehicles, which was the main goal of the presented work.

**Comment 19:** Page 7: why does '14' follow the per mil value?

**Response 19:** Thanks for catching this typo. The 14 was originally a reference to Savard et al., 2017 that did not get converted to the proper citation format in the original SI version. We have updated this citation in the revised SI.

**Reviewer #2**

**Overview:** This study presented very interesting results on the ammonia and ammonium concentrations and isotopic compositions at an urban site, highlighting the importance of urban ammonia emission to the regional atmospheric composition. The year-long observation provides valuable information for evaluating the ammonia emission inventory which is currently under strong debate. In addition, this paper presented detailed observation about the isotopic compositions of both NH3(g) and NH4(p), emphasizing that the isotopic fractionations between these species are significant and highly variable. In general, this manuscript is well written and easy to follow. However, some part of the discussion could use a little bit more clarification. I suggest a minor revision.

**Response:** Thank you for your helpful and constructive feedback. We have addressed the raised points and revised the manuscript according to these suggestions.

**Comment 1:** Lines 246-248: as you mentioned in the previous section, pNH4 is highly correlated with nitrate and sulfate aerosols in the atmosphere, meaning the partitioning between NH4 and NH3 should be mostly controlled by the atmospheric composition. In another word, the amount of pNH4 should be determined by multiple factors, such as the total emission of NHx, NOx and SO2 concentration in the atmosphere, and maybe water content. So, simply based on the observation that pNH4 concentration are similar among these sites, I would not argue "[pNH4+]

may be more regional representative than [NH3] due to its extended atmospheric lifetime relative to NH3".

**Response 1:** Thank you for this comment. We have omitted Lines 246-248 from the original manuscript in the revised version.

**Comment 2:** Figure 1a: for such a small figure, perhaps it is better if you can only show the sites discussed in this figure and remove places like Boston and New Haven as they are not talked about in this paper.

**Response 2:** We appreciate the feedback; however, we respectfully disagree with this suggestion. One significant point we were hoping to draw from this figure was that for our region (New England; US EPA Region 1), $NH_3$ emission densities tend to be highest for the urban regions compared to the rural regions. Thus, having the varying cities on the map can help orient readers to this finding. To draw more attention to this point, we added the following to the revised manuscript on Page 7; Lines 199-201, "This finding was generally consistent with the NEI-14 estimates for New England that tends to show that annual $NH_3$ emission densities were highest for regions near urban locations (Figure 1)".

**Comment 3:** Lines 286-299: I would recommend a more detailed discussion here as it presents interesting results of the isotopic fractionation between the two species, at least moving Figure S2 into the main text and discuss a little bit more about why the seasonal cycle occurred.

**Response 3:** Thank you for the feedback. We have moved Figure S2 into the main text, which is now Figure 6 in the revised manuscript. We have also provided more discussion on the observed $\Delta\delta^{15}N$ values, including the observed seasonality, "There was a strong seasonal $\Delta\delta^{15}N$ pattern with higher values during colder periods, and $\Delta\delta^{15}N$ was weakly correlated with temperature (r = -0.55, p <0.01; Fig. S1) suggesting that these values were difficult to predict. The observed $\Delta\delta^{15}N$ was significantly lower than the expected temperature-dependent theoretical isotopic equilibrium values between $NH_3$ and $NH_4^+$ of 35±3‰ at 25 ºC (Walters et al., 2018) and previous field $\Delta\delta^{15}N$ observations (Savard et al., 2017), indicating that incomplete isotopic equilibrium between $NH_3$ and $pNH_4^+$ was achieved at the study site. This result has important implications for previous $\delta^{15}N$ source apportionment studies of $NH_3$ and $pNH_4^+$, which commonly utilize an assumed and theoretically calculated phase-dependent fractionation (Zhang et al., 2021; Pan et al., 2016; Gu et al., 2022a, b; Berner and Felix, 2020). A potential explanation for the observed incomplete isotopic equilibrium would be that localized $NH_3$ emissions perturbed the isotopic equilibrium between $NH_3$ and $pNH_4^+$, which may take tens of minutes to several hours to be achieved. Indeed, previous laboratory dynamic flow chamber experiments have demonstrated that fresh $NH_3$ emissions tend to result in $\Delta\delta^{15}N$ values below the theoretically predicted value (Kawashima and Ono, 2019). Additionally, there may be other contributing isotope effects between $NH_3$ and $pNH_4^+$ such as the hypothesized kinetic isotope effect associated with $NH_3$ diffusion to an aerosol surface leading to a lower $\delta^{15}N(pNH_4^+)$ value compared to $\delta^{15}N(NH_3)$ (Pan et al., 2016). The observed

$\Delta\delta^{15}N$ seasonality remains difficult to explain. Still, we speculate that it may be related to higher localized emissions of $NH_3$ during warmer periods that perturb the $NH_3/pNH_4^+$ isotope equilibrium and/or seasonal changes in PM chemical compositions such as higher $NH_4NO_3$ during colder months". This update was made on Pages 9-10, Line 259-275 in the revised manuscript.

**Comment 4:** I would argue against the role of R4 in explaining the variable isotopic fractionation. NH3 has a very high pKa (9.26), and aerosol water is usually acidic (at least pH<8), so in aqueous aerosols, NH3(aq) is almost non-existent, almost 100% NH4+. So, only R2 and R3 should be important, and the equilibrium fractionation factor should be somewhere between 31 to 34 permil. The lower isotopic fractionation in the summer may be more related to 1) lower isotopic fractionation factor observed by Kawashima and Ono (2019), and 2) a more important role of kinetic isotopic fractionation.

**Response 4:** Thanks for this suggestion. In our discussion of $\Delta\delta^{15}N$, we have removed $NH_{3(aq)}/NH_{3(g)}$ exchange as a possible driver of the observed variabilities and focus instead on $NH_3/NH_4^+$ equilibrium, diffusion isotope effect, and disequilibrium due to fresh $NH_3$ emissions. The changes to the revised manuscript were documented in response to Comment 3. Further, since we no longer needed a reference to $NH_{3(aq)}/NH_{3(g)}$ exchange, we simplified the introduction by omitting lines 66-101 from the original manuscript.

**Comment 5:** Line 400: discussing how the relative contribution change by season without considering the change in concentration can be misleading. For example, the relative contribution of vehicle emission is lower in the summer and higher in the winter – however once you fold in the concentrations of NHx, you can see the contribution of vehicle emission (in ug/m3) may not change that much. I would recommend the authors revise Figure 8 to include the concentration information, so it is clear for the readers to see the variations of each source, as a sanity check.

**Response 5:** Thank you for this suggestion, and we agree that displaying only the relative fraction of our source apportionment results could be misleading. Thus, in the revised version, we have updated Figure 8 (now Figure 9 in the revised manuscript) also to include the mass-weighted $NH_x$ source apportionment. We can now more clearly see that from a mass-contribution perspective, vehicle emissions were consistent throughout our yearly record despite significant seasonal relative contribution changes. Both volatilization and industry emission have strong seasonality for the relative fractional and mass-weighted contributions. We have discussed the mass-weighted contributions in the revised manuscript on Pages 13-14, Lines 400-405, "The annual and seasonal mass-weighted contributions of the considered sources were calculated utilizing the $NH_x$ concentrations (Figure 9B). Overall, vehicles tended to be a consistent source of urban $NH_x$ with contributions of 35.2±2.6, 33.3±3.5, 32.8±5.3, 35.2±4.3, and 35.4±4.8 nmol/m$^3$ for the annual mean, spring, summer, autumn, and winter, respectively. Mass-weighted contributions for both volatilization and fuel combustion follow their relative fractional profiles with significant seasonal patterns that peaked during summer compared to winter". Furthermore, in this discussion, we

clarified the mixing model results as either relative fractional or mass-weighted contributions. Interestingly, this calculation demonstrates that the mass-weighted contribution of vehicles was relatively consistent throughout the year, which we discuss in context with the NEI predictions in our response to Comment 6.

**Comment 6:** In addition, using the source appointment results, the authors should discuss how it is different from NEI2014. It is clear that our observation showed strong discrepancy vs. emission inventory. For example, emission inventory suggests higher emission in the winter, considering lower mixing height winter, the NHx concentration should be significantly higher than summer, but we see the opposite trend. Isotopic evidence should at least tell us if the seasonal trends of the sources in the emission inventory agree with our observations.

**Response 6:** Thank you for this suggestion, and we have attempted to place our mixing model source apportionment results into context with the predicted NEI as much as possible. First, we acknowledge that it can be difficult to make a quantitative comparison between our source apportionment results and the NEI-14, because the NEI is at a county-level resolution. While our results suggest that our measurements might be more reflective of the local region, long-range transport was also identified as an important contributor to $NH_x$ at our study site. Thus, while the measured $NH_3$ seasonal concentration profile did not match with the SMOKE output for Providence County, it did match with the more rural locations within New England that show a higher emission during summer compared to winter. Further, it is unclear if our one study site would reflect the NEI that represents the entirety of Providence County. Thus, in the revised manuscript, we pointed out that this comparison can be difficult, "The source apportionment results were compared with the predicted $NH_3$ emissions from the NEI-14. We acknowledge that this comparison may not yield quantitative results because the NEI-14 was at a county-level resolution and our single study site may not represent all of the county-level $NH_3$ emission prediction; however, this comparison yields a qualitative understanding in uncertainties of urban $NH_x$" on Page 14; Line 412-415 in the revised manuscript.

Further, we have compared our vehicle and fuel combustion source apportionment results with the NEI-14, "Overall, the seasonally consistent mass-weighted contribution of vehicle emissions source apportionment results was consistent with the NEI-14 that predicts nearly uniform vehicle emissions throughout the year (Figure 4). However, the NEI-14 predicts a lower contribution of annual vehicle emissions in our study location of 31.9% compared to our mixing model results (46.8±3.5%). Our mixing model source apportionment results indicate a relatively low fractional and mass-weighted contribution for stationary fuel combustion for winter, despite the NEI-14 indicating that residential fuel (natural gas and oil) combustion was the largest emission source of $NH_3$ at our study site, the rural CASTNET cites, and other cities with significant heating demands (Zhou et al., 2019). While we acknowledge that the stationary fuel combustion $\delta^{15}N(NH_3)$ emission signatures were uncertain, the mixing model and seasonal [$NH_3$] results would suggest that residential $NH_3$ emissions were overpredicted in the NEI-14, while vehicle emissions may be underpredicted. Thus, vehicle and fuel combustion emission factors may need to be revisited to more accurately model urban [$NH_3$] and predict its human and ecological impacts" on Page 14; Lines 415-418 in the revised manuscript.

---

## Author Response (AR2)

We think the editor for their helpful comments on our revised manuscript. As recommended by the editor, we have included a statement about the lack of evidence for the importance of biomass burning $NH_x$ contributions to our study site in Providence, RI, and expanded on our discussion of the EPA National Emission Inventory residential wood combustion emission estimates. We also supported that biomass burning was not a main source of $NH_3$/$pNH_4^+$ from correlation plots between the measured $NH_3$ and $pNH_4^+$ with potassium ion data from a nearby Chemical Speciation Network. This analysis indicated that $K^+$ was weakly correlated with $NH_3$ and $pNH_4^+$ and was included as an additional figure in the Supplement (Figure S3). Lastly, we mentioned that excluding biomass burning emissions as well as other miscellaneous sources of urban $NH_x$ does not impact the goal of our mixing model results, which is to identify the temporal contributions of the main identified sources at our study site location.

These changes and additions were made on Pages 12-13; Lines 369-380 in the revised manuscript, "Biomass burning, while a significant global source of $NH_3$ (Behera et al., 2013), was not considered in the mixing model since there was insufficient evidence from the local wind direction and long-range transport analysis that it was a major contributing source to our study location. Further, the NEI-14 predicted residential wood combustion represented less than 5% of the annual emission of $NH_3$ in Providence County, with seasonal variation, including higher relative emissions during the colder months (Figure 4). Still, potassium ($K^+$), a common biomass burning tracer, from $PM_{2.5}$ samples collected from the nearby CSN site in East Providence, RI, was not significantly correlated with $NH_3$ (r=0.019; p= 0.857) and weakly correlated with $pNH_4^+$ (r = 0.233; p = 0.022) excluding an outlier on July 4th (Figure S3). We acknowledge that there are additional miscellaneous $NH_3$ sources in an urban environment, including pets, household products, and humans (Ampollini et al., 2019; Sutton et al., 2000; Li et al., 2020); however, we assumed that these sources were negligible compared to the main identified emission sources. Excluding biomass burning and other miscellaneous sources of $NH_3$ was not expected to impact the goal of the mixing model calculations, which was to estimate the relative amounts of the main identified $NH_3$ emission sources and their temporal variation at the Providence, RI study site."